# The Role of the Periodic Table of the Elements of Green and Sustainable Chemistry in a High School Educational Context

Carlos Alberto da Silva Júnior [1,2,*] , Carla Morais [3,*] , Dosil Pereira de Jesus [2] and Gildo Girotto Júnior [2]

1 Department of Chemistry, Federal Institute of Paraiba, Sousa 58805-345, Brazil
2 Institute of Chemistry, University of Campinas, Campinas 13083-970, Brazil; dosil@unicamp.br (D.P.d.J.); ggirotto@unicamp.br (G.G.J.)
3 Department of Chemistry and Biochemistry, University of Porto, Porto 4099-002, Portugal
* Correspondence: carlos.alberto@ifpb.edu.br (C.A.d.S.J.); cmorais@fc.up.pt (C.M.)

**Abstract:** The Periodic Table of the Elements of Green and Sustainable Chemistry (PT-GSC) represents a potentially meaningful tool for teaching and learning Green Chemistry. However, there is a lack of studies exploring the application of the PT-GSC in educational contexts. To contribute to filling this gap, a qualitative and participant approach was developed to examine the effects of using the PT-GSC in a high school setting, with a focus on analyzing the associated challenges and opportunities. Over a five-week period, 23 high school students enrolled in a chemistry course at a public school in Brazil worked in small groups to develop solutions for a case study addressing socio-scientific issues related to water scarcity in the local region using elements from the PT-GSC. Results from both the pre- and post-questionnaires, along with the written case study resolutions, provide evidence of the students' knowledge gains, particularly in critical scientific literacy for Green and Sustainable Chemistry Education. The findings showed that the PT-GSC is an interdisciplinary tool for introducing students to Green Chemistry concepts within the broader societal and scientific ecosystem. The implementation of novel case studies incorporating elements from the PT-GSC is a way to support our ongoing work with students and the public, contributing to a sustainable future.

**Keywords:** green chemistry; high school; case study; education for sustainable development (ESD)

## 1. Introduction

Green and Sustainable Chemistry Education (GSCE) is a growing multidisciplinary field that provides environmentally sustainable practices in the teaching and learning process [1,2]. Historically, the concepts of Green Chemistry began to be defined in the last three decades [3,4]. Its significance lies in shaping individuals into critical and reflective citizens who are not only well versed in chemistry principles but also understand the broader implications of their actions on the environment.

Recent publications in this field highlight the dedication of educators to promoting sustainability awareness through GSCE initiatives. There are contributions to teacher training [5–8] and curricula [9,10], offering valuable insight into the incorporation of sustainable principles into chemistry. However, the adoption of anachronistic teaching practices (the traditional model of education), the gradual reduction of investments, and the lack of educational materials are some barriers that hinder the broader dissemination of this field [11].

In Brazil, it has been observed that the organization and implementation of GSCE initiatives are still limited [12–14]. While there is no previous research on the incorporation of GSCE in Brazilian high schools, recent research has focused on assessing its presence in national teacher training programs [12,14]. The findings reveal a scarcity of GSCE. Only 23% of educational institutions offering chemistry courses have subjects on Green Chemistry [14]. This observation leads us to consider that the scenario in high school education may be similar. According to Marques et al. (2023, p. 18) [15], "Green Chemistry

Education in the country has not yet reached maturity, still being in the development phase". The scarcity of educators with expertise in GSCE makes it challenging to integrate this theme into school curricula. Teachers share knowledge based on their personal learning experiences and through training [16].

It is important to highlight that GSCE extends beyond environmental sustainability and encompasses the other pillars of sustainability, including economic and social settings. In this scenario, it is indispensable that GSCE can enhance environmental education. Therefore, the low frequency of this teaching in curricula becomes concerning as it weakens studies in research, education, and outreach for sustainable development. Without teachers capable of adequately addressing Green Chemistry in the classroom, environmental education, in general, is compromised. It is essential to allocate resources and enhance the educational framework for Green and Sustainable Chemistry across its diverse facets.

According to the literature, authors have proposed models for the integration of Green Chemistry into chemistry curricula [17–19]. In Brazil, Sandri and Santi Filho (2019) [19] expanded upon the frameworks proposed by Burmeister, Rauch, and Eilks (2012) [20], classifying the contributions of teaching Green Chemistry in three models. In Model 1, teachers explore Green Chemistry through practical experimentation, involving aspects such as solvent selection scree or scale reduction. Students are primarily tasked with executing prescribed procedures without a systematic and discerning analysis of chemical sustainability. Model 2 integrates Green Chemistry principles with established concepts in chemistry, emphasizing the importance of discussing and implementing metrics such as green star [21] or life cycle assessment. This model encourages active student participation in designing experimental protocols and engaging in discourse about sustainability issues related to the chemical content. In Model 3, students are critical and intentionally engaging in complex discussions about the relationships between science, technology, and society (STS).

Considering the analyzed knowledge gaps and the imperative need to incorporate GSCE, it is important to seek elements for the construction and assessment of teaching proposals based on this field. However, as Marcelino et al. (2023) alert, it is essential to avoid approaches that are limited to isolated events or are poorly evaluated. One possibility that arises in this context is the use of systemic productions on GSCE to underpin the construction of proposals.

In 2019, researchers Anastas and Zimmerman created the Periodic Table of the Elements of Green and Sustainable Chemistry (PT-GSC) [22]. With this innovative and didactic tool, the authors introduced the concept of "elements of Green Chemistry" or "figurative elements of PT-GSC" aimed at fostering a sustainable future [17,22]. It is important to note that these novel elements differ from the chemical elements traditionally found in the periodic table of the elements. Instead, the proposed elements by Anastas and Zimmerman (2019) are primarily focused on providing social, economic, political, moral, and ethical guidelines [23].

The PT-GSC presents a comprehensive arrangement of 90 "figurative elements", structured into 7 horizontal rows and 18 vertical columns or groups, which are further classified into 4 distinct sets known as "blocks". The "humanitarian elements" block comprises 13 "figurative elements", while the "Green Chemistry and Green Engineering elements" block encompasses 40 elements distributed across 10 groups. Similarly, the "enabling system conditions elements" block contains 30 figurative elements organized into 5 groups. Lastly, the "noble elements" block consists of 7 figurative elements.

This succinct overview of the PT–GSC aims to elucidate some of its fundamental characteristics. As with the historical evolution of the periodic table of the elements, it is envisaged that proposals for refinement and enhancement of the PT-GSC will continue to emerge. As articulated by Anastas and Zimmerman (2019), the PT-GSC remains an evolving framework, open to the incorporation of new "figurative elements" in the future. Hence, it is evident that this alternative periodic table remains susceptible to examination and possible revisions.

Application of the PT-GSC can be integrated with didactic models [24] or teaching methodologies [25,26]. An effective approach, especially when incorporating socio-scientific issues, is the case study (CS) method. Case studies introduce students to real-world problems, emphasizing self-directed learning [27–29]. This approach puts the student at the center of the educational process, encouraging thorough understanding and critical engagement with the subject matter. Furthermore, adoption of the perspective encompassing the three pedagogical moments—problematization, knowledge organization, and knowledge application—can significantly contribute to the effective sequencing of teaching [30]. This structured approach ensures a logical progression in guiding students through the learning process.

In this context, our research questions are: (a) What are the impacts on the learning experience of high school students when employing the PT-GSC in the context of GSCE? (b) What are the challenges and opportunities that are related to this alternative periodic table?

To date, there is no research that specifically focuses on using the PT-GSC in educational contexts, which underscores the pioneering nature of our study, to the best of our understanding. Therefore, this investigation will tackle this issue. The main aims of this study are twofold. The first aim is to examine the impacts on the learning experience of final-year high school students when employing the PT-GSC as a didactic and interdisciplinary resource within the framework of a case study, specifically in the context of GSCE. The second aim is to identify the challenges and opportunities related to this alternative periodic table. In doing so, the innovative approach of using the PT-GSC in an educational context is focused on understanding the essential criteria for future applications.

The subsequent sections of that work are structured as follows: the materials and methods used to teach the PT-GSC and to assess the student's knowledge and perceptions. The third section presents results, which were distributed in five subtopics. The fourth section was dedicated to discussing our findings and practical implications, also presenting limitations. Finally, this study concludes with highlights and future recommendations.

## 2. Materials and Methods

### 2.1. Pedagogical Approach and Participants

For this research, a participant approach was employed to examine the effects of implementing the PT-GSC in a high school setting. Specifically, the perspective of the three pedagogical moments (problematization, knowledge organization, and knowledge application) proposed by Delizoicov et al. (2021) [30] and the framework recommended by Sandri and Santi Filho (2019) [19] were adopted. With this approach, the PT-GSC was introduced as a didactic and interdisciplinary resource in the education of 23 high school students enrolled in an integrated technical course in environment at the Federal Institute of Education, Science, and Technology of Paraiba (IFPB) Campus Sousa, Brazil. All participants engaged voluntarily, providing their consent through informed assent/consent forms. The study was conducted in accordance with the Declaration of Helsinki and approved by the Human Research Ethics Committees at the University of Campinas (protocol code 72002223.3.0000.5404, approved on 19 September 2023) and at the Federal Institute of Paraiba (protocol code 72002223.3.3001.5185, approved on 4 October 2023).

### 2.2. Instruments

Regarding the instruments used in this study, we analyzed the data using both quantitative and qualitative methods. Specifically, quantitative analysis was employed for the ASK-GCP questionnaire to numerically discern trends among the participants. Conversely, qualitative analysis was utilized for data derived from the resolutions of the case study, aiming to comprehend the interpretations and insights derived from the responses. This analysis will be explained more in the following sections.

### 2.2.1. Questionnaires

Table 1 summarizes the goals and question types in the pre- and post-tests. All tests were administered individually, ensuring participant anonymity. The pre-test included 3 open-ended questions; 3 closed-ended questions with the response options "yes", "no", or "maybe"; and 9 questions on a 5-point Likert scale with the options "strongly agree", "agree", "neutral", "disagree", and "strongly disagree" (Appendix A); and 12 true/false questions adapted from recent literature [31] (Appendix B), which are further explained in the next paragraph. In the post-test, these same 12 true/false questions were readministered, along with 15 questions on a 5-point Likert scale (Appendix C) and 3 open-ended questions.

**Table 1.** Goals and types of questions in pre- and post-tests.

| Tests | Types of Questions | Goals |
|---|---|---|
| Pre-test $N = 27$ | Open-ended ($N = 3$) Closed-ended ($N = 3$) Likert scale ($N = 9$) True–false ($N = 12$) [1] | Assess the class's learning strategies and perceptions regarding the discipline of chemistry and evaluate the student's knowledge of Green Chemistry. |
| Post-test $N = 30$ | Open-ended ($N = 3$) True–false ($N = 12$) [1] Likert scale ($N = 15$) | Assess the class's thoughts on research methods and general questions and evaluate the student's knowledge of Green Chemistry. |

[1] Adapted from Grieger, Schiro and Leontyev (2022) [31].

The pre- and post-tests were applied to evaluate the students' knowledge of Green Chemistry. This test contained 12 true–false items, and it was based on the Assessment of Student Knowledge of Green Chemistry Principles (ASK-GCP) developed by Grieger et al. (2022) [31]. The ASK-GCP protocol was adapted to align with the specific contextual nuances of a Brazilian high school environment. The exclusion of statements deemed inappropriate for students at this academic level was undertaken, coupled with the adjustment of select statements to consider the specificity of the local reality. A notable illustration of this refinement is exemplified by the reformulation of the statement from "Ethanol derived from corn is an example of a biomass chemical" to "Ethanol derived from sugar cane is an example of a biomass chemical". This modification was necessary because Brazil assumes a preeminent role as the global leader in sugar cane production [32], providing clear evidence of the prevailing dominance in ethanol production from this tropical grass within the national context. Appendix B presents the statements of the ASK-GCP.

In addition to this true–false instrument, the aim was to assess students in a more qualitative manner. For this purpose, the case study (CS) method [27–29] was employed to analyze the texts generated by the class.

### 2.2.2. Case Study (CS) Method

The case study (CS) method has the potential to enhance an active and meaningful learning process in schools and universities [27–29]. As a derivative of the problem-based learning (PBL) model, the cases represent either real-life scenarios or fictional narratives. Their purpose is to kindle students' motivation and active engagement, prompting them to take charge of decision making in everyday and complex situations, such as socio-scientific issues [28,29]. In the teaching of chemistry, this method has been applied to various topics, including carbohydrates [33], water pollution [34,35], and the greenhouse effect [36]. It is even utilized in the instruction of Green Chemistry for deaf students [37–40]. These examples underscore the fertile ground of the CS method from an inclusive, critical, and transformative perspective.

According to the literature [28], our case study was classified as a mixed-method because students were divided into small groups (up to 5 students) who prepared a discussion for the class. Bernardi and Pazinato (2022) highlighted that this hybrid form is the most used in chemistry education. Our case study was used to facilitate reflection, discussion,

and arguments [41] within a critical perspective of Green Chemistry education [19]. In more detail, the narrative addressed the issue of water scarcity in the city of Sousa, Brazil, prompting the formation of a team led by a professor named Juliana (fictitious name), along with local stakeholders. Divergent perspectives on the causes, encompassing distribution and public awareness of sustainable water usage, emerged during discussions. Professor Juliana proposed integrating the PT-GSC into the research. The team, committed to a systematic approach, resolved to delve deeper into the PT-GSC and present a comprehensive solution to the mayor.

Besides the CS, each group received a printed PT-GSC, five source texts on water scarcity, and was instructed to write an argumentative text and also watch a documentary about the local water crisis. They were free to add additional information, not directly drawn from the source texts, documentary, or the PT-GSC. Data analysis was conducted on the final texts produced and presented by the class, and a total of five written pieces were evaluated.

### 2.3. Procedure

This research was conducted over the course of five sessions, with each session representing three chemistry classes with a duration of 50 min each. In total, there were 15 classes with the students. Next, the procedures undertaken in each session will be outlined. The study began with the presentation of the research to the class and the administration of the informed assent/consent forms. All participants chose to participate voluntarily. Subsequently, a survey with 27 questions was administered individually and anonymously to assess the students' prior knowledge, perceptions regarding the discipline of chemistry, and learning strategies. Additionally, the case study was read aloud to the class, and finally, the class was divided into five groups for further discussion. This first session was identified as the initial problem discussion.

From the second to the fourth sessions, dialogical and participative classes were conducted, guided by the three pedagogical moments. These classes focused on the principles of Green Chemistry [3], the elements of the PT-GSC [17,22], and the conscientious use of water [42]. The use of the PT-GSC was emphasized in both planning and conducting classes. Before each session, PT-GSC elements were explored to enhance the understanding of Green Chemistry. It was consistently integrated into class content. For example, in the organic reactions revision class, the teaching of the addition reaction highlighted "atom economy", "environmental factor", and "additive synthesis", which are figurative elements of the PT-GSC. Mahaffy's four levels of chemical representation were also explored [43–45].

In the last session, students presented their case study solutions, marking the knowledge application phase proposed by Delizoicov et al. (2021) [30]. The narrative addressed the issue of water scarcity in the city where the high school is located. Each group shared their written output related to the PT-GSC, discussed challenges, and received feedback from the research team acting as mediators. The class was organized in a circle to encourage collaboration. The learners submitted their written productions after the group presentation. Subsequently, they took a post-test individually and anonymously, consisting of 12 closed-ended and 3 open-ended questions to assess their knowledge. Additionally, a final questionnaire with 15 items was given to gather students' perceptions about the research's methodological paths, totaling 30 questions. Figure 1 shows the procedural steps in this research.

### 2.4. Data Analysis

For the analysis of the ASK-GCP in the pre- and post-test conditions, the mean accuracy rate was assessed. Furthermore, the paired *t*-test for the means of two paired samples was employed to calculate the *p*-value. In hypothesis testing, the *p*-value signifies the probability of making an error in accepting the alternative hypothesis (H1).

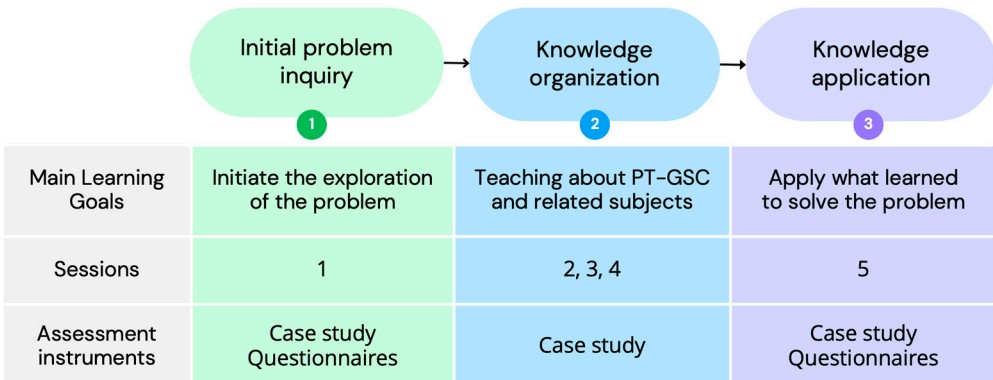

**Figure 1.** The procedural steps in this research are based on the three pedagogical moments (initial problem inquiry, knowledge organization, and knowledge application), describing the main learning goals, sessions, and assessment instruments.

The five final texts produced and presented by the class were analyzed, resulting in the evaluation of a total of five written pieces. The segmentation into units of analysis (UAs) was executed by the first author. Following the approach of Scheuer et al. (2014) [46] and Souza and Queiroz (2018) [47], each UA was considered as a statement ending with a period or semicolon. This allowed for assessing the students' level of written production. Following Souza and Queiroz (2018), our analysis focused on the total number of segmented UAs without a qualitative evaluation of the statements. Concerning the quality of content, the type of evidence (personal or authoritative) and the nature (scientific, social, environmental, commercial, political, economic, cultural, health, and ethical) of each UA were examined. In the literature, this qualitative analysis has been widely used to categorize students' texts [48–50]. Finally, the figurative elements of the PT-GSC used in the texts were identified. The collected data were transformed into graphs and/or figures with the assistance of tools such as Excel and PowerPoint. The generation of word clouds was facilitated through the utilization of the freely accessible tool (https://www.jasondavies.com/wordcloud/) (accessed on 31 January 2024).

*2.5. Research Questions*

The research aims to address two primary questions:

(a) What are the impacts on the learning experience of final-year high school students when the PT-GSC is employed as a didactic and interdisciplinary resource within the framework of a case study, particularly in the context of GSCE?

(b) What challenges and opportunities are associated with this alternative periodic table?

In examining these questions, the focus is on understanding the effects of utilizing the PT-GSC in an educational setting and identifying the challenges and opportunities it presents. This approach aims to elucidate the essential criteria for future applications of the PT-GSC in education.

## 3. Results

*3.1. General Perceptions Regarding Learning Strategies*

The survey with a 5-point Likert scale (Appendix A) was part of the pre-test. Figure 2 displays the responses for items 1–3. It is noticed that most students agreed on using their learning in practice (53.4% agreed), although one-third were neutral regarding this aspect. Similarly, the students involved in this analysis demonstrated a preference for group work, with 46.7% strongly agreeing and 13.3% agreeing, yet they did not exhibit a similar inclination toward consuming articles and content in the general media, with 46.7% expressing neutrality and 20.0% disagreeing.

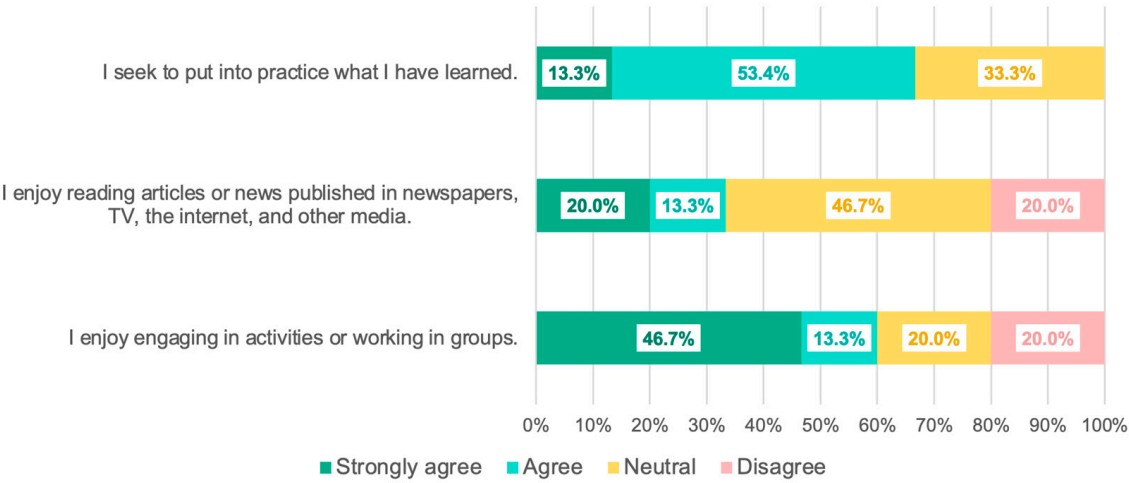

**Figure 2.** Responses for items 1–3 on a 5-point Likert scale.

Figure 3 displays responses for items 4–6 (Appendix A). Learners were asked about their interest in discussing local social and environmental issues, showing a range of opinions. Additionally, they were queried about the effectiveness of activities in their environmental studies course, with approximately 40.0% strongly agreeing, 26.7% agreeing, and 33.3% being neutral. Another question explored the relevance of chemistry to daily life, revealing differing opinions on its practical importance.

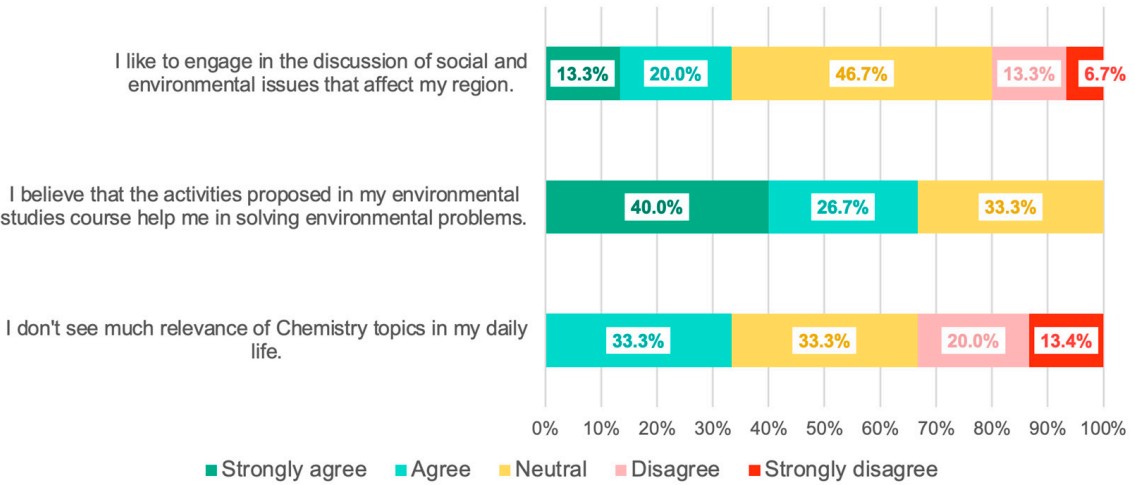

**Figure 3.** Responses for items 4–6 on a 5-point Likert scale.

Figure 4 illustrates the responses for items 7–9 (Appendix A), investigating students' perceptions of three key areas: reading scientific articles, writing argumentative texts, and studying chemistry. The results highlight the varied opinions among students. While some strongly agreed or agreed with reading scientific articles, others were neutral or disagreed. Similarly, opinions on writing argumentative texts, such as the essay in the National High School Exam for university admissions in Brazil, varied, with some enjoying it and others not. Regarding the study of chemistry, students' opinions varied, with individuals expressing agreement, neutrality, or disagreement. Overall, the findings emphasize the diverse attitudes students maintain toward these academic activities.

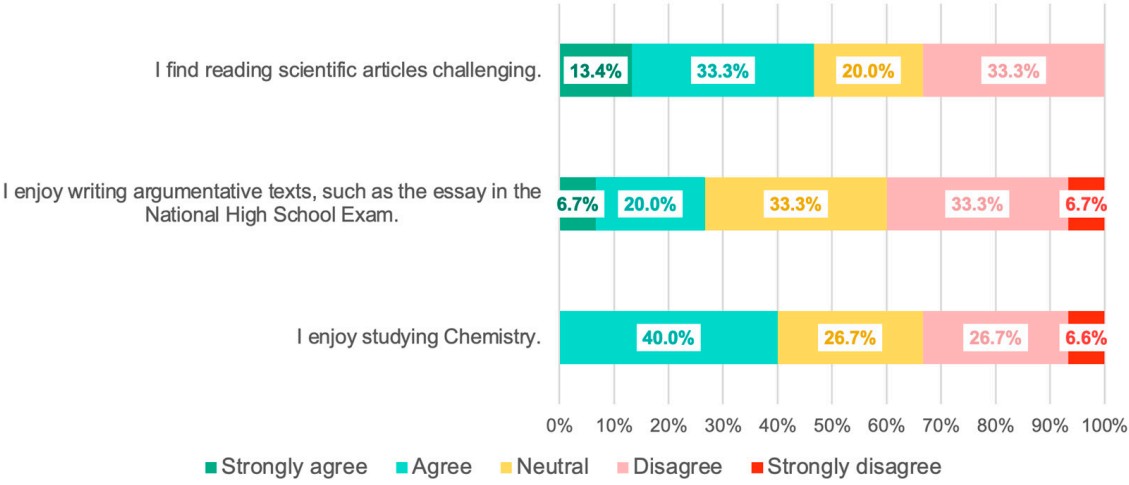

**Figure 4.** Responses for items 7–9 on a 5-point Likert scale.

### 3.2. Prior Knowledge of Research Terms or Expressions

In Figure 5, the results for the three closed-ended questions in the preliminary survey questionnaire are observed, which contain the options "yes", "no", and "maybe". The objective was to assess the class's prior knowledge regarding some terms or expressions relevant to the research. Figure 5A indicates that 80% of the class was familiar with "Green Chemistry", suggesting prior exposure to this concept. However, it is worth noting that despite this prior exposure, their initial definitions of Green Chemistry were vague. Most students lacked awareness of the PT-GSC and the Sustainable Development Goals (SDGs), as indicated in Figures 5B and 5C, respectively. These findings underscore differences in students' knowledge levels, reflecting varying degrees of exposure to different concepts. Despite their familiarity with the term "Green Chemistry", understanding of the PT-GSC and SDGs appeared limited.

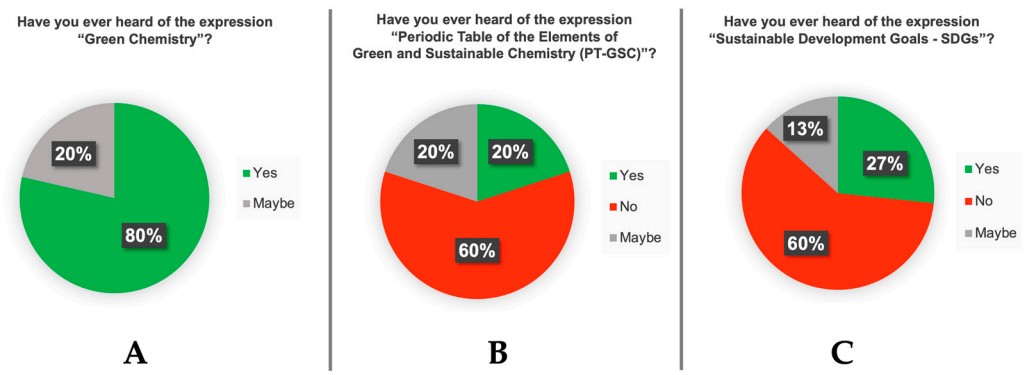

**Figure 5.** Responses to the closed-ended questions: (**A**) "Have you ever heard of the expression: Green Chemistry?", (**B**) "Have you ever heard of the Periodic Table of the Elements of Green and Sustainable Chemistry (PT-GSC)?", (**C**) "Have you ever heard of the expression: "Sustainable Development Goals—SDGs?".

Figure 6 shows word clouds displaying the most repeated words in responses to the question "*What do you think the Periodic Table of the Elements of Green and Sustainable Chemistry (PT-GSC) is?*". It is noted that the most frequently cited term was "elements", as the majority of the class presumed that the PT-GSC would consist of chemical elements. Presented below are several examples of responses: "*It would resemble a chemical table, but with a focus on elements related to the environment*", "*A table exclusively housing elements pertaining to the environment and sustainability*", and "*A table featuring chemical elements*".

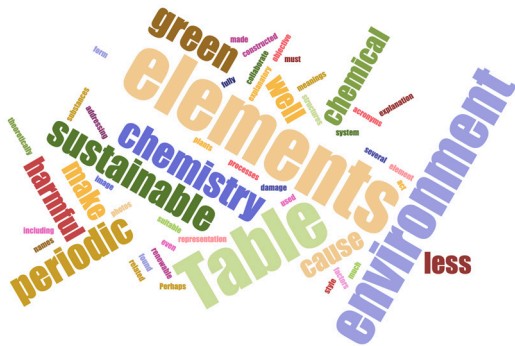

**Figure 6.** Responses to the question in the pre-test, "What do you think the Periodic Table of the Elements of Green and Sustainable Chemistry (PT-GSC) is?".

*3.3. Knowledge in Pre- and Post-Tests*

Figure 7 shows word clouds displaying the most repeated words in responses to the question "*What do you think Green Chemistry is?*" for both the pre-test and post-test. It is observed that there was a predominance in both sets of results of terms related to Green Chemistry, such as "*sustainable*" and "*environmental*". However, the students' understanding of what Green Chemistry is differed before and after the research. The terms "*sustainable*" and "*environmental*" in the pre-test were used very vaguely and without precise responses (Figure 7A). For example, some students wrote: "*It would be more sustainable chemistry, focused on the environment*", "*Chemistry that studies the environment and its forms of benefits and harms through the environment*", and "*Subjects related to chemistry but involving the environment and sustainable issues*". There were also confusing responses such as "*Chemistry that deals with food*" and "*It must be part of chemistry that studies and analyzes the chemical interactions between plants and their relationship with nature*". In the post-test, there were responses more aligned with the standard concept of chemistry [3]. One student wrote: "*Green chemistry is a more sustainable alternative that aims to reduce waste and environmental pollution, reduce energy consumption, toxicities, and other types of waste generation*". Another student responded: "*Chemistry that helps reduce environmental problems and impacts*". Yet another affirmed: "*It is the chemistry that aims to reduce by-products and thus decrease the impacts generated in nature*".

**Pre-test** **Post-test**

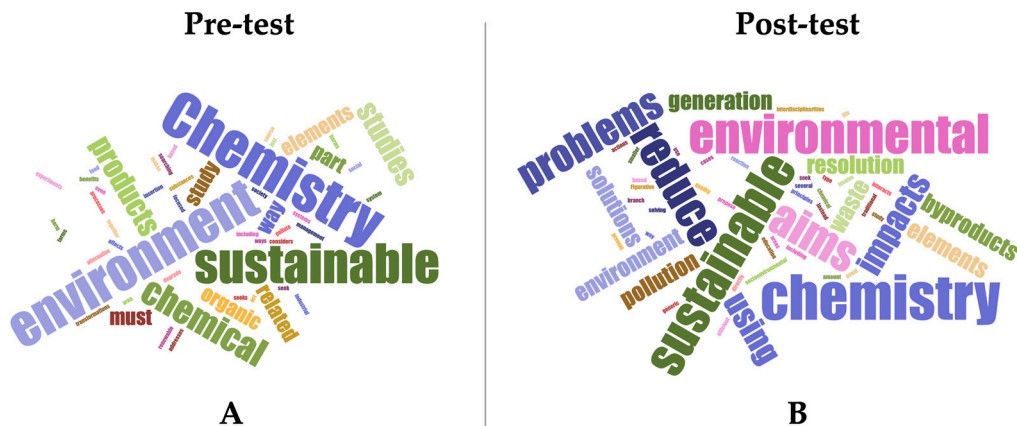

**A** **B**

**Figure 7.** Responses in word clouds to the question: "What do you think Green Chemistry is?" in the (**A**) pre- and (**B**) post-tests.

Figure 8 shows word clouds displaying the most repeated words in responses to the question "*What is the first principle of Green Chemistry?*" for both the pre-test and post-test. In the pre-test (Figure 8A), no students mentioned "*prevention*" in their response, and six scholars admittedly did not know the first principle of Green Chemistry. Some students had conceptual misunderstandings, such as prioritizing the study of plants or studying foods.

For example, one student wrote: "*Prioritize the study of plants, their chemical peculiarities*". Another learner, in turn, answered, "*the use of natural products*". In the post-test (Figure 8B), the majority correctly answered the question, with "*prevention*" highlighted. Other terms in the word cloud resulted from longer responses. For example, one student emphasized prevention as taking measures to reduce damage. Only one student left the question blank in the post-test.

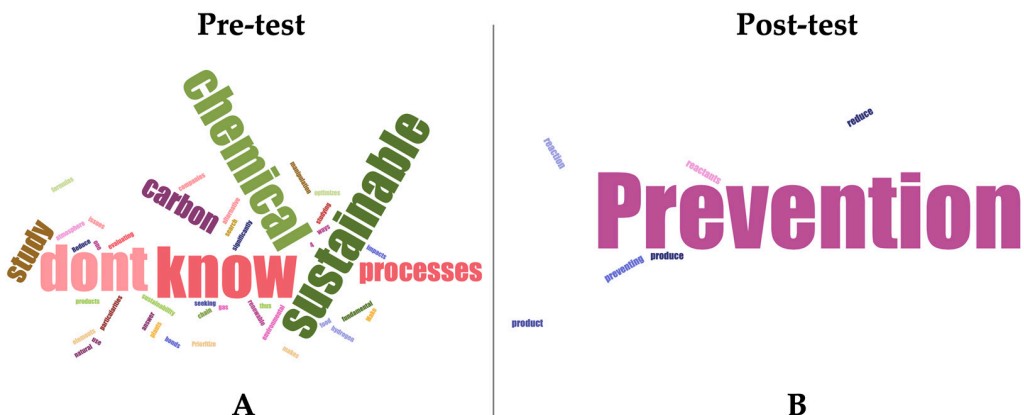

**Figure 8.** Responses in word clouds to the question: "What is the first principle of Green Chemistry?" in the (**A**) pre- and (**B**) post-tests.

Analyzing Figure 9, the results of the ASK-GCP obtained in the pre- and post-research applications are compared. It is noted that the average percentage of correct responses increased from 45.0% to 75.0%. The significance test indicated a genuine, non-random difference, as evidenced by $p < 0.05$. This means that there was less than a 5% chance that the observed difference occurred purely by chance. Further elucidation on the results of the *t*-test can be found in Appendix D.

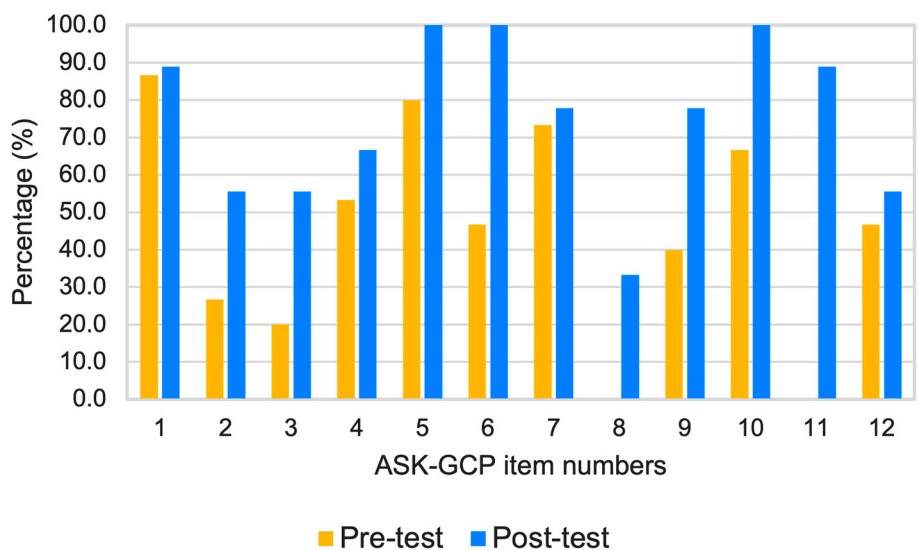

**Figure 9.** Percentage of correct responses for the ASK-GCP in the pre- and post-tests.

The improvement in items related to the reduction of exposure to hazards (item 5), designing reactions with fewer byproducts (item 6), real-time monitoring of the process (item 10) is highlighted, with the percentage of correct answers reaching 100%. These results provide evidence for the effectiveness of gains in students' knowledge.

### 3.4. Written Productions

The following results are based on analyzing five texts written by student groups intended to provide solutions for the case study. The number of units of analysis (UAs), the type of evidence (personal or authoritative), the nature of arguments (scientific, social, environmental, commercial, political, economic, cultural, health, ethical), and the incorporation of figurative elements from the PT-GSC were examined. Table 2 shows the number of units of analysis (UAs) for each team. In total, 82 UAs were coded, with an average of approximately 16 UAs per written text.

**Table 2.** Number of Units of Analysis (UAs) per group of students.

| Group | Number of Units of Analysis (UAs) |
| :---: | :---: |
| G1 | 15 |
| G2 | 14 |
| G3 | 28 |
| G4 | 17 |
| G5 | 8 |

Regarding the analysis of sources of evidence, all teams presented some type of evidence in their argumentative texts. A higher frequency of authoritative sources of evidence was observed. Regarding the analysis of the nature of each UA, it was possible to identify some of the types of arguments listed by Souza and Queiroz (2018) [47].

Table 3 shows the figurative elements of the PT-GSC used for each student group. In total, five elements were cited. Groups G1 and G3 cited three elements each, while groups G2 and G4 mentioned two elements each. Group G5 did not explicitly present any elements.

**Table 3.** Elements of the PT-GSC per group of students.

| Group | Elements of the PT-GSC | Numbers and Symbols for Each Element |
| :---: | :--- | :---: |
| G1 | - zero waste.<br>- capital investment.<br>- extraordinary chemical knowledge comes with extraordinary responsibility. | 36 Z   86 Ci   90 K |
| G2 | - access to safer and reliable water.<br>- zero waste. | 11 Sw   36 Z |
| G3 | - access to safer and reliable water.<br>- alternatives assessment.<br>- capital investment. | 11 Sw   17 Aa   86 Ci |
| G4 | - access to safer and reliable water.<br>- capital investment. | 11 Sw   86 Ci |

The top-cited elements were "access to safer and reliable water" (n = 3) and "capital investment" (n = 3). "Zero waste" was mentioned twice, while "extraordinary chemical knowledge comes with extraordinary responsibility" and "alternatives assessment" were each cited once. Regarding the organization in four blocks of PT-GSC, students mentioned "humanitarian elements", "enabling system conditions elements", and "noble elements". However, they did not mention elements from the block called "Green Chemistry and Green Engineering elements".

Table 4 shows excerpts from the texts written by student groups, with particular attention drawn to the figurative elements referenced in the respective texts. Groups G1, G2, and G3 demonstrated diligence in referencing the figurative elements by specifying their respective symbols. Conversely, group G4 made direct references without employing this symbolic representation.

**Table 4.** Excerpts written per group of students.

| Group | Excerpts Written and Number of the Units of Analysis | Mentioned Elements |
|---|---|---|
| G1 | According to the Periodic Table of the Elements of Green and Sustainable Chemistry, financial investment, represented by Ci, implies that investing could lead to quality management, thereby eliminating a lack of oversight and improving services for the population. (UA6-G1) | 86 Ci |
| G2 | So, it is important to analyze how the figurative elements Z (zero waste) and Sw (access to safe and reliable water) can help in addressing such issues. (UA3-G2) | 11 Sw    36 Z |
| G3 | In this case, we can analyze that there is no assurance of access to water reliably (Sw), as residents allege that the water comes with a cloudy coloration and unpleasant odors, deviating from the standards established in the regulation. (UA12-G3) | 11 Sw |
| G4 | The national water resources policy posits that water is a public good. Therefore, financial investment in the field of treated water not only represents an opportunity for financial returns but also plays a fundamental role in promoting access to clean water, environmental preservation, and the advancement of sustainable practices. (UA14-G4) | 86 Ci |

It is important to note that although group G5 did not mention any figurative elements in their text, they did suggest a solution for the case study. However, the text was brief and lacked references.

### 3.5. Final Perceptions Regarding Research Methods

Figure 10 presents the responses for items 1–15 on a 5-point Likert scale in the final assessment (Appendix C). The analysis of students' responses indicates positive outcomes from incorporating the PT-GSC in GSCE. For example, 77.8% strongly agreed, while 22.2% agreed that "*Attending classes improved my understanding of Green Chemistry*". This alignment in percentages was also noted in the statement "*Attending classes made me think about using water more consciously in my daily life*". Positive outcomes extended to group work, where students felt a sense of responsibility and citizenship. Recognizing the appropriateness of time spent on cases and the importance of team members contributed to a positive learning experience. Concerning "*I believe that the teacher's role in organizing and evaluating the classes was important*", 88.9% strongly agreed and 11.1% agreed.

Moreover, students showed increased awareness of social and environmental issues, promoting responsible consumption locally and globally. They expressed a desire for more integration of the PT-GSC across subjects and increased discussions on Green Chemistry. For the statement "*I believe that Green Chemistry classes helped me become more aware of social and environmental issues and encouraged responsible consumption locally and globally*", 55.6% strongly agreed and 44.4% agreed. The unanimous agreement on sustainable development as a shared mission and interest in participating in future Green Chemistry projects

highlights the lasting impact of this educational approach on students' perspectives and involvement in sustainability initiatives.

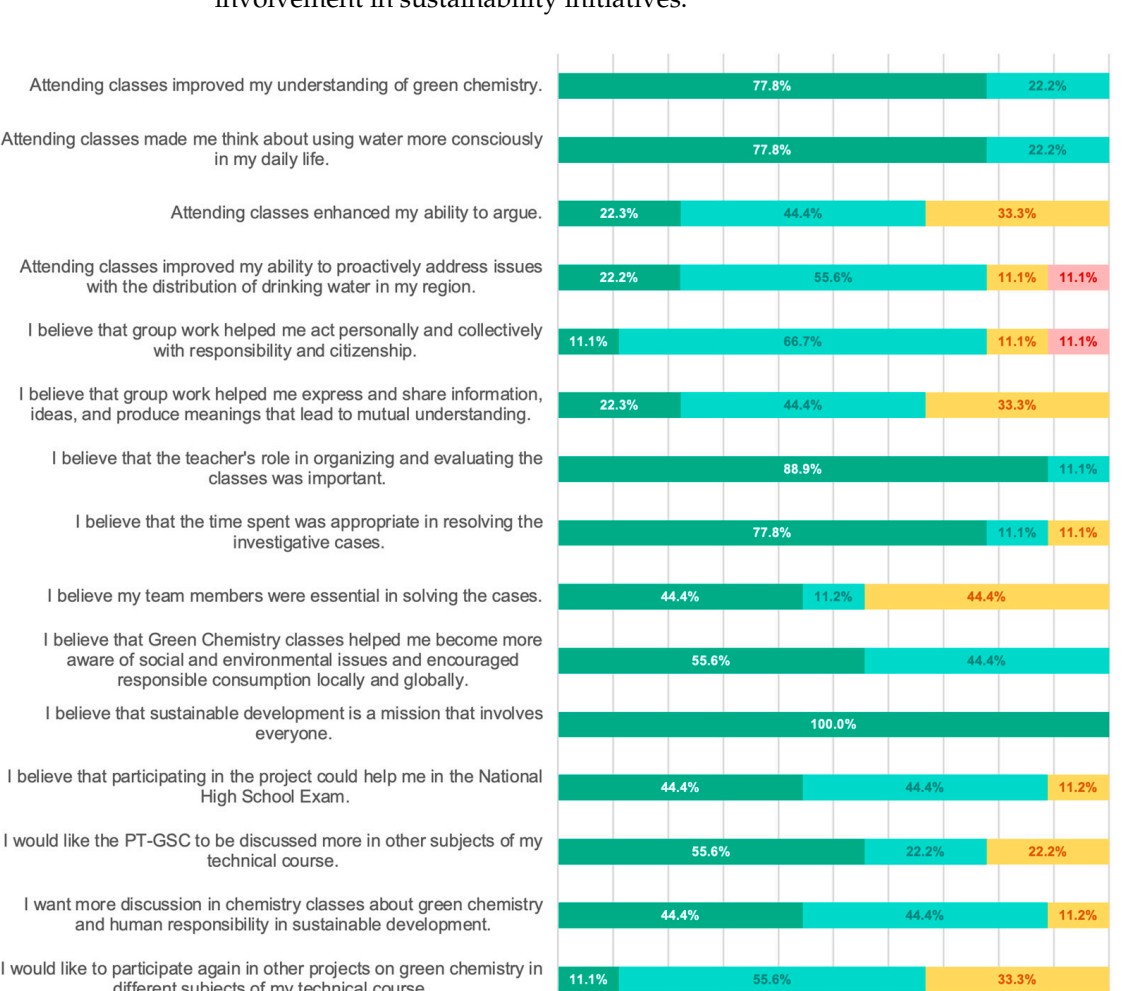

**Figure 10.** Responses for items 1–15 on a 5-point Likert scale in the final assessment.

## 4. Discussion

### *4.1. PT-GSC in an Educational Context*

As previously stated, the purpose of this investigation was to conduct a thorough examination of the impacts on the learning experience of high school students when utilizing the PT-GSC as a teaching and interdisciplinary resource in the context of a case study. Furthermore, the aim was to identify any challenges and opportunities that may arise with the use of this alternative periodic table and present research data to address these two central objectives.

4.1.1. Impacts on the Learning Experience of Students: Moving from Conceptual to Critical Scientific Literacy Level

Overall, our approach with the PT-GSC showed better student responses in relation to the content covered, as seen in the analysis of open-ended questions in both the pre-test and post-test responses. In defining Green Chemistry, the pre-test results indicated an unclear understanding among students, with the vague use of terms like "*sustainable*" and "*environmental*" (Figure 7A). However, the post-test results demonstrated improved clarity, with learners expressing a more accurate understanding (Figure 7B). Similarly, in identifying the first principle of Green Chemistry, the pre-test had conceptual misunderstandings

(Figure 8A), but the post-test showed improvement, emphasizing "*prevention*" (Figure 8B). These results highlight our approach's impact in fostering a deeper and clearer understanding among students, with open-ended questions allowing for a nuanced exploration of their comprehension.

In the literature, there are reports of tests assessing high school students' initial perceptions [51,52]. The results show similarities, with occasional misconceptions like linking Green Chemistry to plant studies [51]. Therefore, it is crucial to explore students' prior understanding for clearer explanations during classes. In a study involving undergraduates from the College of Chemistry at UC Berkeley, Armstrong et al. (2024) [53] used tests to track changes in students' grasp of Green Chemistry and see how their existing "green" knowledge affected learning. The researchers concluded that learners demonstrated an improved understanding after completing a course on Green Chemistry. Furthermore, the authors asserted, "(...) high school classes are critical points of entry and intervention to build normative Green Chemistry understanding (...) If Green Chemistry is truly a framework for chemistry, then it should be present in all chemistry education to allow the greatest number of students the opportunity to hear and learn about Green Chemistry concepts and practices" (Armstrong et al., 2024, p. 130) [53].

In the examination of ASK-GCP outcomes, it was observed that the average percentage of correct responses increased from 45.0% to 75.0%. As a formative assessment, the ASK-GCP has been shown to be sensitive to student learning gains. Examining item 11, focusing on the benefits of enzymes, there was an increase from zero to 88.9% accuracy. The literature on high school education covers studies about catalysis [54,55]. Additionally, it was explained that "enzyme" is a figurative element within the PT-GSC, located in the catalysis group, referencing the ninth principle of Green Chemistry [17]. Another positive development was a 33.3% accuracy improvement in understanding the "environmental factor" [56–58]. The class had limited knowledge of Green Chemistry metrics, but they learned about the "environmental factor", recognizing it as a figurative element in the PT-GSC. It is worth noting that the PT-GSC has a category named "metrics", providing an opportunity for further exploration, especially in classes involving experiments. This indicates potential for expanding students' understanding of Green Chemistry principles and metrics in the academic context.

Continuing the discussion on the ASK-GCP, results were also seen in items 5, 6, and 10, achieving 100% accuracy in the post-test. Although not everyone answered all questions correctly, there was an overall 30.0% increase in accuracy in ASK-GCP responses, in accordance with the conceptual level.

Another important point to discuss is how students not only improved their conceptual understanding of Green Chemistry and its principles but also grasped its real-life application. This may be designated as the critical level. According to Andrade and Zuin (2023) [18], there are three levels of scientific literacy for experimentation in Green and Sustainable Chemistry. By analyzing student experiments in a university chemistry course, these authors found that most aligned with conceptual literacy. In our study, experimentation was not conducted, as our objective was to propose a methodology suitable for implementation in schools lacking such facilities, a circumstance still prevalent in countries like Brazil. Consequently, in our discussion, the critical level was understood as involving the adoption of socio-scientific issues that propel chemistry education. This critical approach aligns with Model 3 proposed by Sandri and Santi Filho (2019) [19]. At this juncture, the most integrative role of the PT-GSC was observed, as shown in students' responses to the case study. Figure 11 illustrates how the utilization of different figurative elements of the PT-GSC can contribute to the transition from a conceptual to a critical level through the incorporation of case studies.

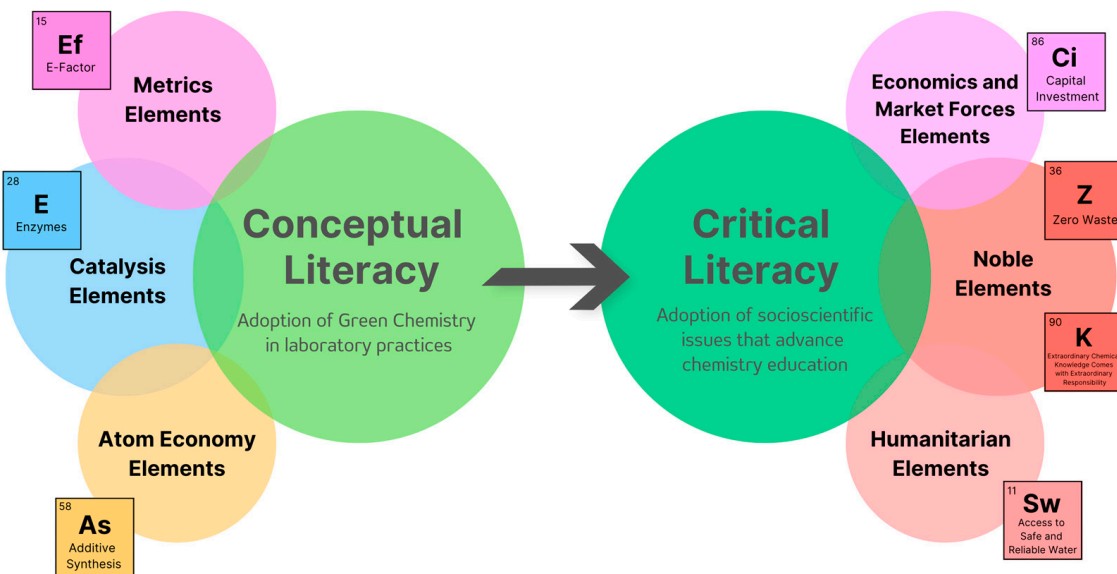

**Figure 11.** Scientific literacy levels for Green and Sustainable Chemistry Education (GSCE), describing the migration from conceptual to critical scientific literacy level of GSCE, using figurative elements from the PT-GSC and case studies.

The illustration presented in Figure 11 also points toward the discussion of assessments that are more critical. It is common for teachers to use multiple choice questions in preparation for national university entrance exams. In Brazil, there are reports of classes that have become true showcases for shortcuts and tricks to ace national tests [59,60]. It is necessary to regain the purpose of education. Students are not being prepared for exams but for life. In this context, the PT-GSC can assist in constructing assessments that are more holistic, or it can be used as an educational assessment tool. In any case, with the PT-GSC, the discussion extends beyond the 12 principles of Green Chemistry proposed by Anastas and Warner [3]. The students' responses highlight a Green Chemistry that is more contextualized.

This contextualization is seen in the case narrative, addressing a local issue of water scarcity. It is important to mention that this issue is quite serious. During the project, classes were suspended for three days due to a lack of water at the school. Positive percentages from Likert scale questions highlight the effectiveness of the project methodology. Students expressed a desire for more Green Chemistry discussions and more frequent use of the case study method.

In the analysis of argumentative production, group G3 stood out with the highest number of units of analysis (UAs), totaling 28, while group G5 had the lowest, with precisely 8 UAs. The increased number of UAs in group G3 can be attributed to the use of the Toulmin argumentative model. Their approach involved using short sentences to explain the fundamental elements of the argumentative structure. Consequently, as UAs were derived from statements ending with periods and/or semicolons, the detailed description of Toulmin's structure by group G3 led to a higher count of UAs. It is important to highlight that this group was the only one to provide such a comprehensive explanation of the Toulmin model.

These findings suggest a positive impact on students' learning outcomes. Consequently, regarding the first research question, it can be inferred that the PT-GSC is a didactic tool with the potential to promote critical scientific literacy level of GSCE.

### 4.1.2. Challenges and Opportunities

When employing the PT-GSC as a didactic tool, it is imperative to establish a coherent plan. Teachers can use practical demonstrations, real-life problems, or case studies to help students understand. Herein, the approach of using the PT-GSC had the additional benefit of making it easier to address the 17 Sustainable Development Goals (SDGs). According to our survey, 60% of the class had never heard of SDGs, which allowed for the opportunity to teach them. This work aligns with the discussion of the Sustainable Development Goals (SDGs), specifically SDG 6 (clean water and sanitation). Our research findings highlight the importance of understating how the use of the PT-GSC can impact high school students in providing solutions for local and complex problems. SDG 6 focuses on ensuring access to water and sanitation for all. Our results suggest that elements from the PT-GSC, such as "access to safer and reliable water" (Sw) and "chemistry for wellness" (Cw), may have significant implications in educational contexts, providing a more holistic perception of Green and Sustainable Chemistry.

The importance of the ASK-GCP was noted in analyzing the progress in learning the principles of Green Chemistry; however, it was inferred that our case study allowed a more holistic assessment of a complex and local problem. The majority using elements from the blocks called "humanitarian elements" and "noble elements" contributes to the observation that the PT-GSC can lead to a more social and political discussion. In the literature, GSC is usually addressed in laboratory experiments and higher education [61–72]. Herein, an approach without a laboratory and in a high school context is provided.

An examination of Table 3, where students explicitly linked figurative elements like "access to safer and reliable water" (Sw) and "capital investment" (Ci) with the case study, provides valuable insight. The references to these elements by three distinct groups indicate a shared recognition of their relevance in the case study context. This connection between figurative elements and practical problem solving enriches the learning experience by connecting everyday aspects with chemical knowledge.

Considering this insight, along with the Likert scale responses (Figure 10) emphasizing students' positive perceptions and heightened awareness, the effectiveness of the PT-GSC in GSCE is highlighted. This alignment with the case study not only shows the practical utility of the PT-GSC but also underscores its role as a teaching tool in Green Chemistry education. This study indicates that integrating the PT-GSC with teaching strategies has the potential to promote Green Chemistry instruction, aligning seamlessly with our research objectives.

This study highlights both the challenges and opportunities of using the PT-GSC in teaching Green Chemistry. The challenges include the need to shift assessment methods and the limited research on the use of the PT-GSC in education. On the other hand, the PT-GSC provides a creative framework for educators and supports the sustainable development agenda. Its interdisciplinary nature makes it adaptable to various educational settings, from high schools to universities. This discussion addresses the second research question of this study.

Traditional chemistry has boosted knowledge and technology but struggles with social, complex, and global issues like the water scarcity crisis. Chemists often overlook environmental impacts, and education neglects lab safety, hindering innovation for broader sustainability problems. Working with "noble elements" and "humanitarian elements" may prepare students to address holistic issues by showing how chemistry can be related to society. Students need to learn more than the 12 principles of Green Chemistry; they need to have a more open vision and apply their knowledge in context. As Anastas affirms, "ideally, the future of Green Chemistry is that the term Green Chemistry goes away, because it's simply the way we do chemistry".

### 4.2. Limitations and Future Research

This paper examined the effects of using the PT-GSC in a high school setting, using a 12-item true–false instrument and the case study (CS) method to analyze the effectiveness of gains in students' knowledge. The presented methodology can serve as a reference for

introducing students to Green Chemistry concepts within the broader societal and scientific ecosystem. However, this paper has some limitations. Firstly, this study presented a limited number of participants (n = 23), indicating that our findings may not be generalizable. Additionally, robust statistical tests were not conducted due to the small sample size. Secondly, this work did not assess the complexity of arguments written by the class. Therefore, future studies could enhance this investigation and implement a more systemic thinking approach. There is a possibility of using Toulmin's argument pattern for studying science discourse. This would offer more critical insight for the implementation of the PT-GSC in educational contexts. For future research, it is crucial to investigate the use of the PT-GSC in other types of classes, such as laboratory experimental classes. Additionally, it is important to explore whether students exhibit any misconception between the PT-GSC and the periodic table of elements.

## 5. Conclusions

In addressing our research inquiries, the findings underscore significant impacts on the learning experience of high school students through the implementation of the PT-GSC in the realm of GSCE. Over a five-week period, these students enrolled in a chemistry course at a public school in Brazil worked in small groups to develop solutions for a case study addressing socio-scientific issues related to water scarcity in the local region using elements from the PT-GSC. The results from a survey based on a 12-item true–false instrument conducted in pre- and post-conditions provide evidence of the effectiveness of students' knowledge gains. The transition from a conceptual to a critical level of scientific literacy, as illustrated in Figure 11, aligns with the positive outcomes observed in our analyses of open-ended questions and case studies. The hands-on approach of the PT-GSC improved students' understanding of Green and Sustainable Chemistry.

Furthermore, the challenges and opportunities associated with the PT-GSC as a potentially meaningful tool for teaching and learning Green Chemistry were described by the study's outcomes. The challenges include the need for a shift in assessment methodologies from traditional multiple choice questions to more holistic evaluations, aligning with the critical level of scientific literacy. Additionally, the study identifies the limited existing research on the use of the PT-GSC in educational contexts, emphasizing the necessity for further exploration and validation.

On the other hand, the opportunities presented by the PT-GSC are vast. This alternative periodic table provides a creative reference framework for educators, students, teachers, researchers, and other stakeholders on how chemistry can be compatible with and support the 2030 sustainable development agenda. The interdisciplinary nature of the PT-GSC allows for its integration into various educational settings, supporting educators in addressing local and global challenges. This aligns with the study's potential to promote the use of the PT-GSC not only in high school but also in university settings, showcasing its adaptability and relevance across educational levels.

It is crucial to note that this study is pioneering, as, to date, there is no research specifically focusing on using the PT-GSC in educational contexts, to the best of our understanding. Therefore, this study addresses this gap by shedding light on the unique contributions and potential of the PT-GSC in advancing GSCE.

Finally, our research answers the questions posed, revealing the positive impacts of the PT-GSC on high school students' learning experiences in GSCE and shedding light on the challenges and opportunities inherent in adopting this alternative periodic table. The study advocates for the continued exploration and integration of the PT-GSC in educational contexts to further advance the field of GSCE.

**Author Contributions:** Conceptualization, C.A.d.S.J.; methodology, C.A.d.S.J.; validation, C.A.d.S.J.; formal analysis, C.A.d.S.J.; investigation, C.A.d.S.J.; resources, C.A.d.S.J.; data curation, C.A.d.S.J., C.M., D.P.d.J. and G.G.J.; writing—original draft preparation, C.A.d.S.J.; writing—review and editing, C.A.d.S.J., D.P.d.J., C.M. and G.G.J.; visualization, C.A.d.S.J., D.P.d.J., C.M. and G.G.J.; supervision;

D.P.d.J., G.G.J. and C.M.; project administration, C.A.d.S.J., D.P.d.J. and G.G.J. All authors have read and agreed to the published version of the manuscript.

**Funding:** This research received no external funding.

**Institutional Review Board Statement:** The study was conducted in accordance with the Declaration of Helsinki and approved by the Human Research Ethic Committees at the University of Campinas (protocol code 72002223.3.0000.5404, approved on 19 September 2023) and at the Federal Institute of Paraiba (protocol code 72002223.3.3001.5185, approved on 4 October 2023).

**Informed Consent Statement:** Informed consent was obtained from all subjects involved in the study.

**Data Availability Statement:** This paper is part of the PhD research of one of the authors.

**Acknowledgments:** The authors would like to thank the University of Campinas (UNICAMP), the Federal Institute of Paraiba (IFPB), the São Paulo State Research Support Foundation (FAPESP), the Faculty of Sciences of the University of Porto, and the high school students who participated in the research.

**Conflicts of Interest:** The authors declare no conflicts of interest. The funders had no role in the design of the study; in the collection, analyses, or interpretation of data; in the writing of the manuscript; or in the decision to publish the results.

## Appendix A

**Table A1.** Survey assessment on a 5-point Likert scale.

| Items | Statements |
|---|---|
| 1 | I seek to put into practice what I have learned. |
| 2 | I enjoy reading articles or news published in newspapers, TV, the internet, and other media. |
| 3 | I enjoy engaging in activities or working in groups. |
| 4 | I like to engage in the discussion of social and environmental issues that affect my region. |
| 5 | I believe that the activities proposed in my environmental studies course help me in solving environmental problems. |
| 6 | I don't see much relevance of Chemistry topics in my daily life. |
| 7 | I find reading scientific articles challenging. |
| 8 | I enjoy writing argumentative texts, such as the essay in the National High School Exam. |
| 9 | I enjoy studying Chemistry. |

## Appendix B

**Table A2.** ASK-GCP instrument and answers [1].

| Item | Statement | Answer |
|---|---|---|
| Q1 | An understanding of toxicology and environmental chemistry assists in designing safer chemicals | True |
| Q2 | A reaction that has 100% yield will result in a 100% atom economical reaction | False |
| Q3 | Reactions at elevated temperatures should be prioritized over reactions at room temperature | False |
| Q4 | Fossil fuels are a renewable feedstock | False |
| Q5 | Reduction of exposure to hazards is the best way to minimize accidents | True |
| Q6 | Designing reactions with fewer byproducts is a good method of waste prevention | True |
| Q7 | Ethanol derived from sugar cane is an example of a biomass chemical | True |
| Q8 | The environmental factor equals the mass of waste produced in a chemical process | False |
| Q9 | When designing a synthesis, the use of personal protective equipment is sufficient for controlling exposure to hazards | False |
| Q10 | Real-time monitoring of the process helps to avoid incidents caused by side reactions | True |
| Q11 | A disadvantage of enzymes is that they suffer from poor selectivity, thus producing more derivatives | False |
| Q12 | A catalyst lowers the activation energy, which allows for reduced reactions times | True |

[1] Adapted from Grieger, Schiro and Leontyev (2022) [31].

## Appendix C

**Table A3.** Final assessment on a 5-point Likert scale.

| Items | Statements |
|---|---|
| 1 | Attending classes improved my understanding of Green Chemistry. |
| 2 | Attending classes made me think about using water more consciously in my daily life. |
| 3 | Attending classes enhanced my ability to argue. |
| 4 | Attending classes improved my ability to proactively address issues with the distribution of drinking water in my region. |
| 5 | I believe that group work helped me act personally and collectively with responsibility and citizenship. |
| 6 | I believe that group work helped me express and share information, ideas, and produce meanings that lead to mutual understanding. |
| 7 | I believe that the teacher's role in organizing and evaluating the classes was important. |
| 8 | I believe that the time spent was appropriate in resolving the investigative cases. |
| 9 | I believe my team members were essential in solving the cases. |
| 10 | I believe that Green Chemistry classes helped me become more aware of social and environmental issues and encouraged responsible consumption locally and globally. |
| 11 | I believe that sustainable development is a mission that involves everyone. |
| 12 | I believe that participating in the project could help me in the National High School Exam. |
| 13 | I would like the PT-GSC to be discussed more in other subjects of my technical course. |
| 14 | I want more discussion in chemistry classes about Green Chemistry and human responsibility in sustainable development. |
| 15 | I would like to participate again in other projects on Green Chemistry in different subjects of my technical course. |

## Appendix D

**Table A4.** *t*-test: Means of two paired samples.

| Tests | Pre-Test | Post-Test |
|---|---|---|
| Mean | 45.0083 | 75.0167 |
| Variance | 849.5663 | 473.9961 |
| Observations | 12 | 12 |
| Pearson Correlation | 0.5865 | |
| Hypothesized Mean Difference | 0.0000 | |
| Df | 11 | |
| P(T <= t) two-tail | 0.0012 | |

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
