# Peer review of "The Role of the Periodic Table of the Elements of Green and Sustainable Chemistry in a High School Educational Context"

_sustainability, doi:10.3390/su16062504_

Round 1

Reviewer 1 Report

Comments and Suggestions for Authors

This paper presented a research on using the periodic table of the elements of green and sustainable chemistry (PT-GSC) as a tool to improve the effects for green and sustainable chemistry education (GSCE), in a high school chemical class context in Brazil. Considering the limitations mentioned in Sec 4.2, the whole experiments are clearly described and the results are sound. Still some suggestions:

1. Two Table 1 on page 4. For the first Table 1, why N=28 for pre-test, seems 27 in total?

2. Merge the content of Sec 2.2.1.1 into Sec 2.2.1 and move the second Table 1 to Sec Appendix.

3. As PT-GSC is the key tool for the whole experiments, I would suggest a little bit more introductions of such tool for readers not familiar with it, and for better understanding of the results in Table 3 & 4 and Fig 11.

4. Explicitly answering the RQs and aims mentioned in page 3.

Other mistakes, e.g. Ln 119 to discussing.

Comments on the Quality of English Language

Moderate editing of English language required.

Author Response

Thank you very much for taking the time to review this manuscript. Please find the detailed responses below and the corresponding revisions/corrections highlighted/in track changes in the re-submitted file.

Reviewer comment:

1. Two Table 1 on page 4. For the first Table 1, why N=28 for pre-test, seems 27 in total?

Modification: Agree. The value has been corrected. In this case, the total number of questions on the pre-test is equal to 27.

Reviewer comment:

2. Merge the content of Sec 2.2.1.1 into Sec 2.2.1 and move the second Table 1 to Sec Appendix.

Modification: Agree. The content of Section 2.2.1.1 and Section 2.2.1 has been merged, and Table 1 has been moved to Appendix B.

Reviewer comment:

3. As PT-GSC is the key tool for the whole experiments, I would suggest a little bit more introductions of such tool for readers not familiar with it, and for better understanding of the results in Table 3 & 4 and Fig 11.

Modification: Agree. As the PT-GSC serves as the primary tool for all experiments, two new paragraphs have been included in the introduction to cater to readers who may not be acquainted with it, thereby facilitating a better understanding of the results.

Reviewer comment:

4. Explicitly answering the RQs and aims mentioned in page 3. Other mistakes, e.g. Ln 119 to discussing.

Modification: Agree. In order to provide explicit responses to the RQs and objectives outlined on page 3, paragraphs have been incorporated into the Discussion section, alongside the creation of topics 4.1.1 and 4.1.2.

Reviewer 2 Report

Comments and Suggestions for Authors

The article is well-organized and presents interesting results

I have only a few comments for improvement:

-     ---     Write specific Research Questions RQs. And also specify in the results or discussions section, which results answer each RQ. That way the article is easier to follow.

-   --       Figure 10.  It would better to present a boxplot. Those bars graphs are difficult to derive direct conclusions from.  If you use boxplots you can talk about the median for the ítems.

-     --     It is also recomended to make statistical hypothesis tests for the medians or means as evidence for the RQs. You can talk directly in POST vs PRE results comparison

Author Response

Thank you very much for taking the time to review this manuscript. Please find the detailed responses below and the corresponding revisions/corrections highlighted/in track changes in the re-submitted file.

Reviewer comment:

Write specific Research Questions RQs. And also specify in the results or discussions section, which results answer each RQ. That way the article is easier to follow.

Modification: Agree. Topic 2.5 was introduced to offer a precise description of the RQs. Moreover, within the discussion section, it was clarified which results pertain to each RQ. As a result, sections 4.1.1 and 4.1.2 were incorporated into the Discussion, along with the inclusion of new paragraphs.

Reviewer comment:

Figure 10.  It would better to present a boxplot. Those bars graphs are difficult to derive direct conclusions from.  If you use boxplots you can talk about the median for the items.

Modification: When the results of Figure 10 were represented in a box plot, the legibility of the data was compromised. Consequently, the decision was made to retain the bar graph format, with an emphasis on percentages to enhance clarity. Within the domain of Chemical Education literature, bar graphs with color highlighting are commonly favored for Likert-scale questionnaires. Additionally, the paragraphs are structured to accentuate the principal findings depicted in Figure 10.

Reviewer comment:

It is also recommended to make statistical hypothesis tests for the medians or means as evidence for the RQs. You can talk directly in POST vs PRE results comparison.

Modification: Agree. A statistical hypothesis test was conducted on the means to substantiate the RQs. The ASK-GCP questionnaire was chosen for this analysis because its items were administered both before and after the research, enabling a direct comparison of post- versus pre-results. However, we abstained from conducting this test on other individual questions due to their inherent disparities. For example, all Likert-scale questions in the pre-test differ from those in the post-test.

Reviewer 3 Report

Comments and Suggestions for Authors

The manuscript has got an empirical character, it is divided into the chapters and subchapters typical for this kind of studies. The text is written in relatively understanding form. Authors used scientific language. I have got some comments, if they will be incorporated, the manuscript will be on higher level. The comments are presented below.

1. At the end of theoretical part the authors should present research aims, in the clear form. Because, on the basis of this, the results part would be presented in the more accurate form and as a reviewer I will be able to evaluate the discussion part of the manuscript.

2. The methodology part of the manuscript is unclear for me. I am not able to distinguish, which approach is used, if qualitative or quantitative, because authors wrote about qualitative approach toward obtaining and analyzing data, but according reading of the text I am not sure, if this is truth statement. Because authors used questionnaire and there is a hint toward pre-experimental or quasi-experimental design. This part of the text should ne revised.

3. The using of percentage is not so valuable, authors could use statistical techniques typical for small sample size. In this form, it is not sufficient.

4. As I wrote in the previous lines, I am not able to review the discussion chapter.

5. Some technical comments:

- please use passive in the whole text,

- used References should be only in English language.

I hope my comments are helpful

Author Response

Thank you very much for taking the time to review this manuscript. Please find the detailed responses below and the corresponding revisions/corrections highlighted/in track changes in the re-submitted file.

Reviewer comment:

1. At the end of theoretical part the authors should present research aims, in the clear form. Because, on the basis of this, the results part would be presented in the more accurate form and as a reviewer I will be able to evaluate the discussion part of the manuscript.

Modification: Agree. At the end of theoretical part, topic 2.5 was established to succinctly present the research objectives. Additionally, within this section, the research questions (RQs) were introduced to facilitate comprehension of the subsequent discussion in the manuscript.

Reviewer comment:

2. The methodology part of the manuscript is unclear for me. I am not able to distinguish, which approach is used, if qualitative or quantitative, because authors wrote about qualitative approach toward obtaining and analyzing data, but according reading of the text I am not sure, if this is truth statement. Because authors used questionnaire and there is a hint toward pre-experimental or quasi-experimental design. This part of the text should be revised.

Modification: Agree. The main modifications in the methodological section of the manuscript involve clarifying the approach used. Specifically, a paragraph was added to explain that a quantitative approach was employed for the questionnaires, involving the collection and analysis of tabulated data. Additionally, it was specified that a qualitative approach was utilized in the case study results, as this method allows for classification based on the qualitative aspect of students’ writing. It is worth noting that this explanation became clearer at the beginning of topic 2.2, which discusses instruments.

Reviewer comment:

3. The using of percentage is not so valuable, authors could use statistical techniques typical for small sample size. In this form, it is not sufficient.

Modification: Agree. A statistical hypothesis test was conducted on the means. The ASK-GCP questionnaire was chosen for this analysis because its items were administered both before and after the research, enabling a direct comparison of post- versus pre-results. However, we abstained from conducting this test on other individual questions due to their inherent disparities. For example, all Likert-scale questions in the pre-test differ from those in the post-test.

Reviewer comment:

4. As I wrote in the previous lines, I am not able to review the discussion chapter.

Modification: We appreciate your feedback regarding the discussion chapter review. Despite this limitation, it's important to note that, in response to the RQs and objectives outlined on page 3, paragraphs have been incorporated into the Discussion section, and topics 4.1.1 and 4.1.2 have been introduced. These refinements were implemented to ensure explicit responses and enhance clarity in presenting the key findings aligned with the research goals.

Reviewer comment:

5. Some technical comments:

- please use passive in the whole text,

- used References should be only in English language.

Modification: Agree. Passive voice was employed consistently throughout the text, and the references cited are exclusively in the English language.